# Robust adaptive optics for localization microscopy deep in complex tissue

Marijn E. Siemons[1], Naomi A. K. Hanemaaijer [1,2], Maarten H. P. Kole [1,2] & Lukas C. Kapitein [1✉]

Single-Molecule Localization Microscopy (SMLM) provides the ability to determine molecular organizations in cells at nanoscale resolution, but in complex biological tissues, where sample-induced aberrations hamper detection and localization, its application remains a challenge. Various adaptive optics approaches have been proposed to overcome these issues, but the exact performance of these methods has not been consistently established. Here we systematically compare the performance of existing methods using both simulations and experiments with standardized samples and find that they often provide limited correction or even introduce additional errors. Careful analysis of the reasons that underlie this limited success enabled us to develop an improved method, termed REALM (Robust and Effective Adaptive Optics in Localization Microscopy), which corrects aberrations of up to 1 rad RMS using 297 frames of blinking molecules to improve single-molecule localization. After its quantitative validation, we demonstrate that REALM enables to resolve the periodic organization of cytoskeletal spectrin of the axon initial segment even at 50 μm depth in brain tissue.

[1] Cell Biology, Neurobiology and Biophysics, Department of Biology, Faculty of Science, Utrecht University, Utrecht, the Netherlands. [2] Department of Axonal Signalling, Netherlands Institute for Neuroscience, Royal Netherlands Academy for Arts and Sciences (KNAW), Amsterdam, the Netherlands. ✉email: l.kapitein@uu.nl

Single-molecule localization microscopy (SMLM)[1–3] enables the investigation of the nanoscale organization of cellular structures through repetitive localization of different sparse subsets of fluorophores. SMLM has provided key insights into the nanoscale organization of molecules in cultured cells, including the discovery of sub-membranous actin-spectrin rings in axons[4,5]. Cultured cells, and neurons in particular, have limitations as model systems, because they lack most of the three-dimensional cellular organization and neurochemical conditions that are present in vivo, including extracellular matrix proteins and nutritional support from glial cells. On the other hand, performing SMLM deep inside tissue or organoids has remained challenging due to sample-induced aberrations. Accurate localization requires sufficient signal to noise and a non-aberrated point-spread-function (PSF)[6–8], both of which are compromised when imaging in biological tissue, in which a range of distinct cellular components cause complex light scattering[9].

One solution is to perform low-depth (~10 μm) imaging in thin sections[6–8]. However, in such experiments most cells within imaging range have been sectioned to some extent, precluding live-cell experiments in such preparations. In contrast, acutely cut >250-μm-thick serial sections from the brain preserve most of the characteristics of the neuronal cytoarchitecture and allow the ex vivo study of electrophysiological properties. This brain slice preparation is one of the most widely used preparations in neuroscience and facilitates structure–function studies of intact and electrically active brain cell microcircuits. For these reasons, there is a clear need to perform optical nanoscopy deep inside these aberrating samples.

An important methodology to overcome tissue-induced aberrations is the use of intensity-based adaptive optics (AO), which uses a deformable mirror in the emission path to compensate for wave-front distortions. The required shape of the mirror can be found by iteratively optimizing the contrast of the image (or guide star when possible)[10]. However, in SMLM the acquisitions are noisy and contain a strongly fluctuating amount of signal photons, rendering traditional approaches unusable. Various SMLM-specific AO-methods have been proposed to overcome these issues[11–13], but the exact performance of these different AO-methods has not been consistently demonstrated and it has remained unclear what level of aberrations these methods can correct and under which conditions.

Here, we systematically compare the performance of existing methods using both simulations and experiments with standardized samples. We find that for realistic total signal and background ratios these methods provide only limited correction or introduce additional errors. Careful analysis of the reasons that underlie this limited success enabled us to develop an improved method, termed Robust and Effective Adaptive Optics in Localization Microscopy (REALM), which corrects aberrations of up to 1 rad RMS using 297 frames of blinking molecules, thereby enabling robust SMLM at 50 μm depth and even up to 80 μm depth when pre-correcting spherical aberration.

## Results

Aberrations alter the point spread function (PSF), decreasing contrast and the spatial frequency support (see Fig. 1a–d). The key for intensity-based AO is to find a relevant metric, a quality measure computed from the acquisitions, in combination with an optimization algorithm to efficiently and robustly optimize the metric[14]. To reduce the effect of strongly fluctuating signal levels of the acquisitions, all existing methods propose a weighted sum of the Fourier transform of the acquisition as metric, while differing in the specific weighting of the spatial frequencies and in normalization (see insert Fig. 1d). We termed these methods 1, 2,

and 3, with metrics M1, M2, and M3, corresponding to Burke et al.[11], Tehrani et al.[12], and Mlodzianoski et al.[13], respectively. Because SMLM acquisitions are comprised of pseudo-random point sources, these methods effectively probe and optimize the magnitude transfer function (MTF). As optimization algorithm to maximize the value of the metrics, Burke et al. use model-based optimization, Tehrani et al. use particle-swarm optimization, and Mlodzianoski et al.[13] use downhill simplex optimization.

In order to systematically compare between different methods, we first sought to establish a standardized sample with tunable signal and background values. Therefore, we used a DNA-PAINT-based sample where molecules transiently bind to the coverslip, mimicking blinking[15] (see Fig. 1a). This sample maintains stable signal and noise levels during the complete experimental sequence of several hours (see Methods section and Supplementary Fig. 1). Using a set-up with a carefully calibrated deformable mirror (see Methods section and Supplementary Figs. 2 and 3) we introduced 25 random aberration configurations of 0.75 rad RMS wave-front error, consisting of random combinations of Zernike modes up to the fourth radial order (excluding piston, tip, tilt, and defocus) and assessed how well the various methods were able to correct these aberrations (see Fig. 1e, f). In order to achieve a realistic signal and background level, we tuned the emitter density and intensity to around 20 emitters per frame and 2500 signal photons per emitter. Transmission brightfield illumination was used to substantially increase the background level to 20 photons per pixel in the 400 × 400 pixel field of view, resulting in a total signal background ratio (SBR) of 0.016 (2500 signal photons × 20 emitters/(20 background photons × 400 × 400 pixels)). In addition, we tested the performance of the methods on simulated data sets (see Methods section for simulation details).

In both the standardized experiments and the simulations, we found that the previously proposed methods were unable to meaningfully correct the aberrated wave-front. For example, method 1 and method 2 increased the aberration level in 100% and 48% of the experimental cases, respectively, whereas method 3 decreased the aberration level only by 20% (0.16 rad RMS) on average. The experimentally achieved corrections deviated to some extent from the simulation results (Fig. 1f), likely due to additional noise sources such as read noise and fixed pattern noise from the camera, which are not included in the simulation. Furthermore, the small initial aberration introduced by the DM limited the experimentally achievable aberration level to 0.2 rad RMS (Supplementary Fig. 1e). Nonetheless, both simulations and experimental data show that none of the three methods robustly achieves proper correction (here taken as a Strehl-ratio of 0.9).

To understand why correction often fails, we next examined how the different metrics depend on noise levels and aberrations. The spatial frequency content of acquisitions in non-aberrated and aberrated conditions revealed that, in both cases, spatial frequencies above 1 NA/λ are dominated by noise (see Fig. 1d). Metric M1 and M2 have the highest weights for these frequencies, which includes a large amount of noise in the metric value. In contrast, M3 only weights the low spatial frequencies (<1 NA/λ), where signal levels are much higher. This makes metric M3 the most robust measure of the MTF and explains why M3 enables consistent correction, albeit without reaching diffraction-limited imaging.

We wondered whether the limited correction obtained using metric M3 could be caused by the use of simplex optimization. The simplex optimization is sensitive to local noise in the parameter space as it only compares two values per optimization parameter. In contrast, model-based optimization iteratively corrects Zernike modes by applying a sequence of biases for each Zernike mode. The metric values for these series of acquisitions are then fitted to a curve (the model or so-called metric curve) to

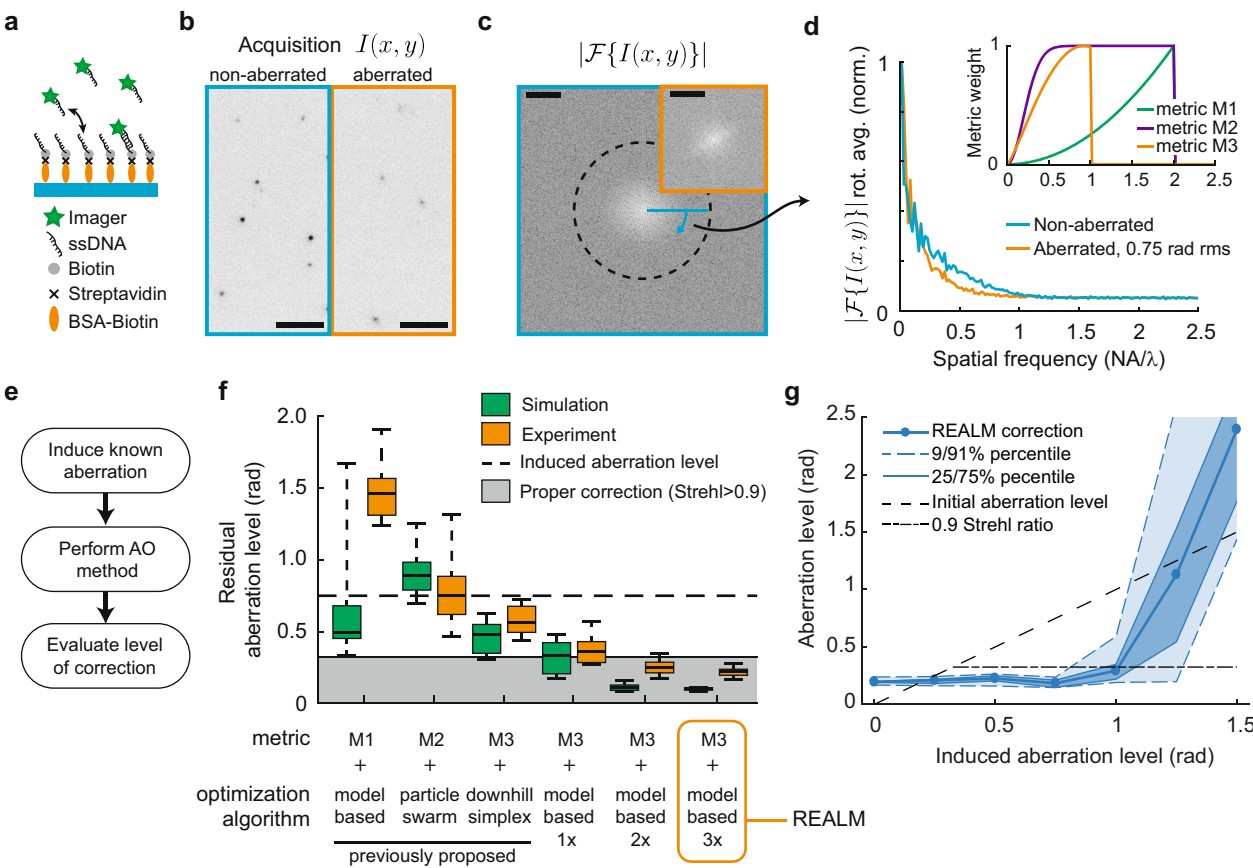

**Fig. 1 Systematic comparison between different AO methods reveals that only REALM achieves robust correction. a** DNA-PAINT test sample: imager strands bind transiently to the coverslip, mimicking single-molecule blinking with consistent signal levels for many hours. **b** Representative acquisition of a non- and 0.75-rad RMS aberrated acquisition ($I(x,y)$) corresponding to **f** ($n = 625$). Scale bar indicates 5 μm. **c** Absolute value of the 2D Fourier transform of a non-aberrated acquisition and aberrated acquisition of **b** (orange insert, center crop) ($|\mathcal{F}\{I(x,y)\}|$). Dashed line indicates 2NA/λ. Scale bars indicate NA/λ. **d** The rotational average of **c** shows that noise dominates spatial frequencies above 1 NA/λ. The major decrease in the MTF due to aberrations occurs between 0.25 and 1 NA/λ. Insert shows the spatial frequency weights of the different proposed metrics. **e** Strategy for comparing AO methods by inducing known aberrations in a well-corrected system. **f** Performance of different AO metrics and optimization algorithms. Metric 3 in combination with model-based optimization leads to robust correction in three correction rounds (297 acquisitions), and we termed this approach REALM. Boxplot indicates 9/91-percentile, 25/75-percentile, and median for 25 random aberration configurations of 0.75 rad rms wave-front error consisting of random combinations of Zernike modes up to the fourth radial order (excluding piston, tip, tilt, and defocus). Each frame (400 × 400 pixels, 26 × 26 μm) contains on average 13 emitters, emitting 2500 photons with a background of 20 photons per pixel. See Supplementary Fig. 1 for more details. **g** Performance of REALM as function of induced aberration level (10 random aberration configurations). Each frame (400 × 400 pixels, 26 × 26 μm) contains on average 17 emitters, emitting 2465 photons with a background of 40 photons per pixel. Source data are available as a source data file.

find the optimum (see Supplementary Fig. 4)[16]. This procedure reduces noise and therefore appears more suitable for AO in SMLM, as originally proposed by Burke et al. To test this, we implemented metric M3 in combination with model-based optimization. We first simulated aberrated acquisitions to assess the metric curve and found that it could be approximated by a Gaussian function with offset within a range of ±1 rad per Zernike mode (see Supplementary Fig. 5). Next, we used a series of simulations to optimize our method in terms of maximum bias range, the number of biases per Zernike mode, and the number of correction rounds (see Supplementary Fig. 6).

These simulations provided two key insights. First, they revealed the importance of varying the bias over a sufficiently large range to confidently estimate the optimal bias (see Supplementary Fig. 6a–c). In order to estimate the optimal bias with a high precision, the contrast in the metric value has to be as large as possible and needs to be probed ~0.5–0.75 rad around the optimum. We found that using a maximum bias range of ±1 rad resulted in the best correction, which is in line with previous work on model-based modal aberration correction[16]. Secondly, these

simulations revealed that the precision of the estimated optimal bias depends on the amount of aberration in the other modes, i.e., the contrast in the obtained metric values for the Zernike mode that is being corrected improves when the overall level of aberration is lower. Consequently, the use of multiple correction rounds improves the correction (see Fig. 1f and Supplementary Fig. 6d–f). The optimization analysis indicated that there are two efficient correction strategies: either 3 correction rounds, with 9 biases per Zernike mode ($3 \times 9 \times 11 = 297$ acquisitions) or 2 correction rounds with 13 biases per Zernike mode ($2 \times 13 \times 11 = 286$ acquisitions). The latter option appeared to perform slightly more robust in the conditions with low SBR, whereas the first option achieved slightly better correction if the aberration profile was very unevenly distributed between Zernike modes. In all cases, it was beneficial to first correct the Zernike modes expected to dominate, such as spherical aberration, because for any mode the correction precision depends on the amount of aberration in the other modes.

We experimentally verified this approach using the DNA-PAINT sample and were able to consistently reduce the induced

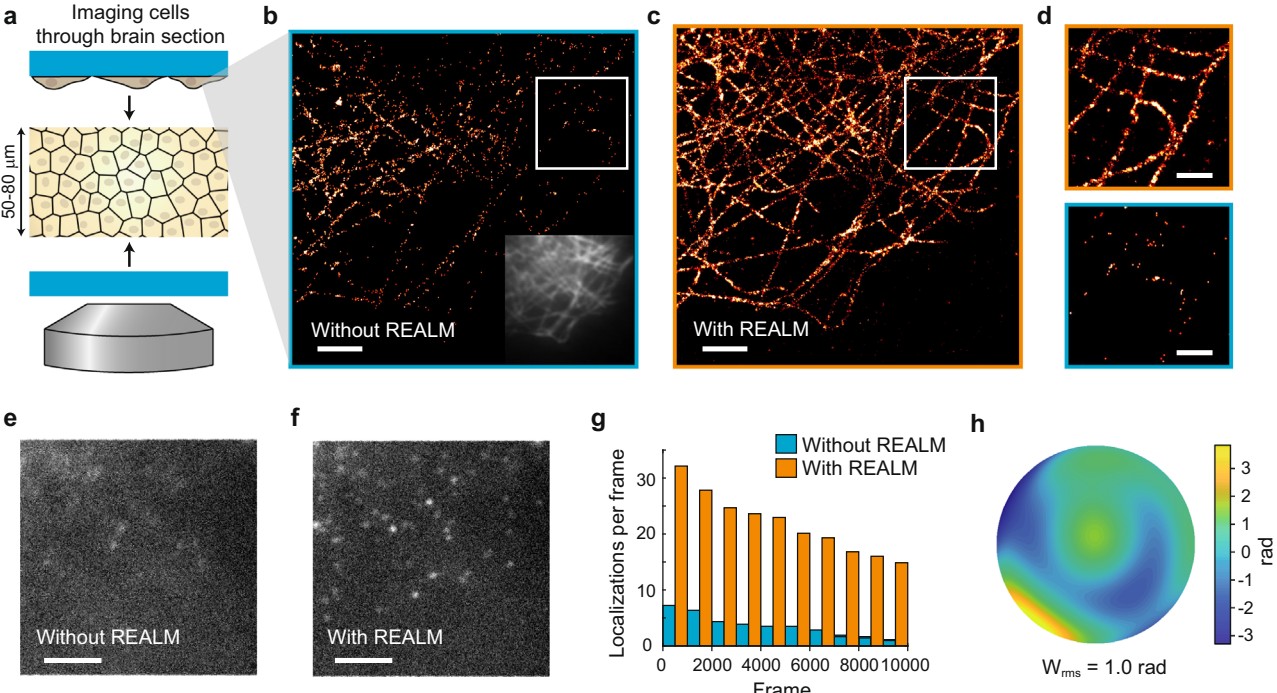

**Fig. 2 Improved single-molecule imaging through brain sections using REALM. a** Illustration of the sample. A 50-μm brain slice is mounted in between two cover glasses, on one of which COS−7 cells were grown. These cells were stained for microtubules (αTub) and imaged through the brain section, to mimic deep-tissue imaging. **b**, **c** SMLM reconstruction of microtubules in COS−7 cells through a 50-μm-thick brain section with 5000 frames without AO sample correction (without REALM, **b**) and with sample-based correction (with REALM, **c**). Insert in panel **b** shows the widefield image prior to correction. FRC resolution is 196 nm and 127 for **b** and **c** respectively. Scale bars indicates 2 μm. **d** zoom of **b** and **c** indicated by white square. Scale bars indicates 1 μm. **e**, **f** Representative single acquisitions corresponding to the reconstruction of **b** and **c** respectively. Scale bars indicate 4 μm. **g** Number of localizations in the acquisition series during which the DM alternated between the system-corrected state (without REALM) and sample-corrected state (with REALM) every 500 frames. **h** Estimated wave-front distortion as estimated using REALM, of which 0.55 rad was used as initial guess for primary spherical aberration. Repeated $n = 10$ times in three distinct samples. Source data are available as a source data file.

wave-front error of 0.75 rad RMS by a factor of two in the first correction round (99 acquisitions). The second correction round (another 99 acquisitions) yielded further improvement and achieved diffraction-limited imaging in 21 out of 25 cases, whereas after the third correction round (297 acquisitions in total) all induced aberrations were corrected (see Fig. 1f). We termed this method Robust and Effective Adaptive Optics in Localization Microscopy (REALM) and implemented it as a freely available and open source Micro-Manager plugin[17] (see Methods section). The performance of REALM was evaluated further by measuring the aberration correction for increasing amounts of induced aberrations and different signal background ratios. This revealed that our method was able to robustly correct aberrations of up to 1 rad RMS (see Fig. 1g and Supplementary Fig. 7) for various signal and noise levels. Even with very low signal levels (6 emitters on average, 1745 signal photons per emitter, and 216 background photons, 0.0003 SBR) REALM was able to correct significant aberrations (see Supplementary Fig. 7c).

We next aimed to test this approach for SMLM using more complex experimental samples. Previously, Mlodzianoski et al.[13] used a water-filled cavity to validate their method. However, such a sample does not fully capture the complexity of biologically realistic aberrations in tissue, where cell bodies, capillaries, and nerve fibers all may act as obstacles for light and distort the wavefront. Others have tested their AO method using fluorescent beads imaged through the nematode C. elegans[12]. While this assay features a richer aberration profile, the beads are bright and do not exhibit blinking dynamics. In an effort to better mimic deep-tissue localization microscopy, we performed SMLM on COS−7 cells stained for microtubules (a staple reconstruction test

in SMLM), but imaged these cells through brain sections of 50 and 80 μm thickness (see Fig. 2 and Supplementary Fig. 8). We chose this sample over stained tissue for our first analyses, because these microtubule immunostainings are highly reproducible and result in recognizable structures. To address the scattering induced by the sample we mounted the brain sections in a glycerol-based buffer instead of a water-based buffer, similar to previous work[6]. This reduces scattering by reducing the local differences in refractive index between the subcellular content and the mounting medium. Slices mounted in this buffer appear almost transparent in regions such as the cortex, indicating that scattering is indeed reduced. We estimated that the refractive index of brain sections in this buffer is around 1.48 (assuming an overall refractive index of 1.4 and a water content of 70% before incubation in the glycerol buffer). We therefore used a 1.49 NA oil objective to ensure larger collection efficiency and less sample-induced spherical aberrations compared to a silicon immersion objective lens (see Supplementary Fig. 9).

With this assay, we performed a systematic comparison between REALM and non-AO conditions by switching between the system-corrected state (without REALM) and the sample-corrected state (with REALM) every 500 frames during the SMLM acquisition (see Supplementary Movie. 1). We found that REALM increased the number of successful localizations (see methods for classification details) up to 6-fold, resulting in a strongly improved reconstruction (see Fig. 2). The resolution estimated using Fourier Ring Correlation (FRC) improved by 35% when using REALM. For the 80-μm-thick slices, we estimated the spherical aberration to be ~100 nm and used this for precorrection. We switched the DM between this pre-corrected

state and the sample-corrected state and again found an improvement in both the number of detected events when using REALM as well as the FRC (24% improvement) (see Supplementary Fig. 8), indicating that non-spherical aberrations contribute considerably to image deterioration as well.

We found that without AO blinking events are detectable only if the events are bright and perfectly in focus. This approach still yielded recognizable features in the reconstruction (see Fig. 2b), but with much lower number of localizations. This was caused by the aberrated PSF, which quickly broadened when molecules were slightly out of focus (±100–200 nm) and therefore significantly hampered detection (see Fig. 2e and Supplementary Movie 1). With REALM, the PSF is more symmetric and remains focused along the focal depth of the objective. This increases the detection of dim blinking events as well as the detection of molecules inside the complete depth of focus. Therefore the reconstruction with REALM also contains microtubule structures that are not visible in the reconstruction without AO (see Fig. 2d). Although the ground-truth aberrations were unknown, the increase in the number of detected events, as measured using the exact same fluorescent structures, demonstrates the success of the AO algorithm.

We next used a similar sample (COS−7 cells imaged through 60 μm thick brain slices) to directly compare REALM to the method proposed by Mlodzianoski et al. (see Supplementary Fig. 10). This revealed that REALM was more effective in restoring the PSF, which consequently improved detection and localization. Overall, this resulted in a 2.2× increase (median) in number of localizations with a localization precision below 20 nm and a 8.6× increase in the number of localizations with a precision below 5 nm. Even when we extended the previously published method by including more Zernike modes than originally proposed, it could not achieve the same correction as REALM. The FRC resolution improved from 150 nm (median) when using the method of Mlodzianoski et al. to 120 nm (median) when using REALM. On average, REALM estimated the aberrations to be ~1 rad RMS, whereas the method of Mlodzianoski et al. resulted in a 3-fold lower estimate, leaving an average residual aberration of ~0.7 rad RMS. These results demonstrate that REALM achieves a 3-fold improvement in aberration correction over previous methods and enables robust single-molecule imaging at 60 μm depth through brain tissue with improved resolution.

Next, we aimed to image structures stained within the tissue itself and focused on the axon initial segment (AIS) of cortical layer 5 pyramidal neurons in rat brain slices of 300 or 400 μm thickness. Landmark SMLM experiments have used neurons cultured on coverslips to reveal that axons display a ~190-nm actin-spectrin based periodic structure called the membrane-associated periodic scaffold[4], which at the AIS includes βIV-spectrin[4,5]. However, due to the limited imaging depth of conventional SMLM, correlating these structures to functional recordings of neurons in brain slices is not possible. In acute brain slices healthy neurons are typically located at >30 μm depth from the slice surface and can be reliably targeted by a patch pipette up to 100 μm depth[18]. Using REALM, we could perform multiplane astigmatic 3D SMLM imaging on βIV-spectrin stained brain sections up to a depth of 50 μm (Fig. 3). We resolved the periodic patterning of this scaffolding protein in 3D and revealed a periodicity of $203 \pm 10$ nm (mean ± s.d., Fig. 3, Supplementary movie 2 and 3, and Supplementary Fig. 11). For details of the estimated aberrations see Supplementary Fig. 12. Beyond 50 μm depth, the increased background levels hindered both REALM and single-molecule detection.

To further examine the improvement achieved by REALM we imaged βIV-spectrin in slices and switched between uncorrected

and corrected mirror states during the acquisition, which resulted in an improved reconstruction upon correction (76 nm FRC with AO, 100 nm FRC without AO, see Supplementary Fig. 13a). In addition to improving the number of detections and the FRC, the use of Adaptive Optics also corrected the loss of ellipticity when performing astigmatic 3D SMLM[13,19]. We observed a depth-dependent increase in aberrations (see Supplementary Fig. 13d) and have shown in previous work[19] that this loss of ellipticity is due to (higher order) spherical aberration, revealing why the applied astigmatism level has to increase when imaging in tissue even with using AO. Without such an increased level of astigmatism, the z-encoding is lost, rendering volumetric multiplane astigmatic 3D SMLM unattainable because individual focal planes cannot be stitched (see Supplementary Fig. 13b, c). Using PSF simulations described in our previous work[19] we could determine the theoretical required astigmatism level at each depth to obtain a constant calibration curve. Application of these estimated levels of astigmatism restored proper z-encoding and allowed volumetric multiplane astigmatic 3D SMLM (Fig. 3, Fig 4 and Supplementary Figs. 11 and 13).

Finally, we also successfully resolved the βIV-spectrin structure of a functionally identified pyramidal neuron in a brain section (Fig. 4). First, electrophysiological recordings were performed on a layer 5 pyramidal neuron (Fig. 4a–d), after which the neuron was filled with biocytin, fixed and stained. After mounting of the brain section and application of the SMLM buffer, the position of the patched neuron was retrieved using the stained biocytin fill and a ×10 objective. Subsequently, SMLM in combination with REALM was used to resolve the nanoscale architecture of βIV-spectrin, demonstrating the possibility to directly correlate functional studies and nanoscopic organization.

## Discussion

In this work, we systematically analyzed the performance of different AO techniques for SMLM using standardized samples. Comparing different methods in an objective and robust manner is challenging, because signal and background levels can vary dramatically between samples and often also rapidly change during acquisitions. To overcome this, we first used DNA-PAINT in combination with a well-calibrated deformable mirror to compare and validate the performance of previously published methods in identical signal and background levels, while systematically varying the level and types of aberrations. We augmented these experiments with simulations to further compare how different methods perform in identical conditions. Furthermore, we imaged densely labeled cells through brain slices of varying thickness, enabling the side-by-side evaluation of different methods in identical conditions with physiological aberrations. Together, these assays revealed that previously proposed AO methods provide only limited correction or in some cases introduced additional aberrations. Careful analysis of the reasons that underlie this limited success enabled us to develop REALM. We demonstrated that REALM can robustly correct aberration levels up to 1 rad RMS in realistic signal and noise levels and using <300 acquisitions. Compared to standard imaging (non-AO) and earlier AO methods, using REALM results in more detected molecules and better FRC resolutions.

Next to these idealized samples, we also tested the performance of REALM in stained brain slices. We demonstrated multiplane astigmatic 3D SMLM imaging on βIV-spectrin stained brain sections up to a depth of 50 μm and resolved the periodic structure in 3D. Importantly, we also resolved this structure in a functionally identified pyramidal neuron, demonstrating the feasibility of directly correlating functional studies to nanoscopic organizations. While imaging βIV-spectrin, typical background

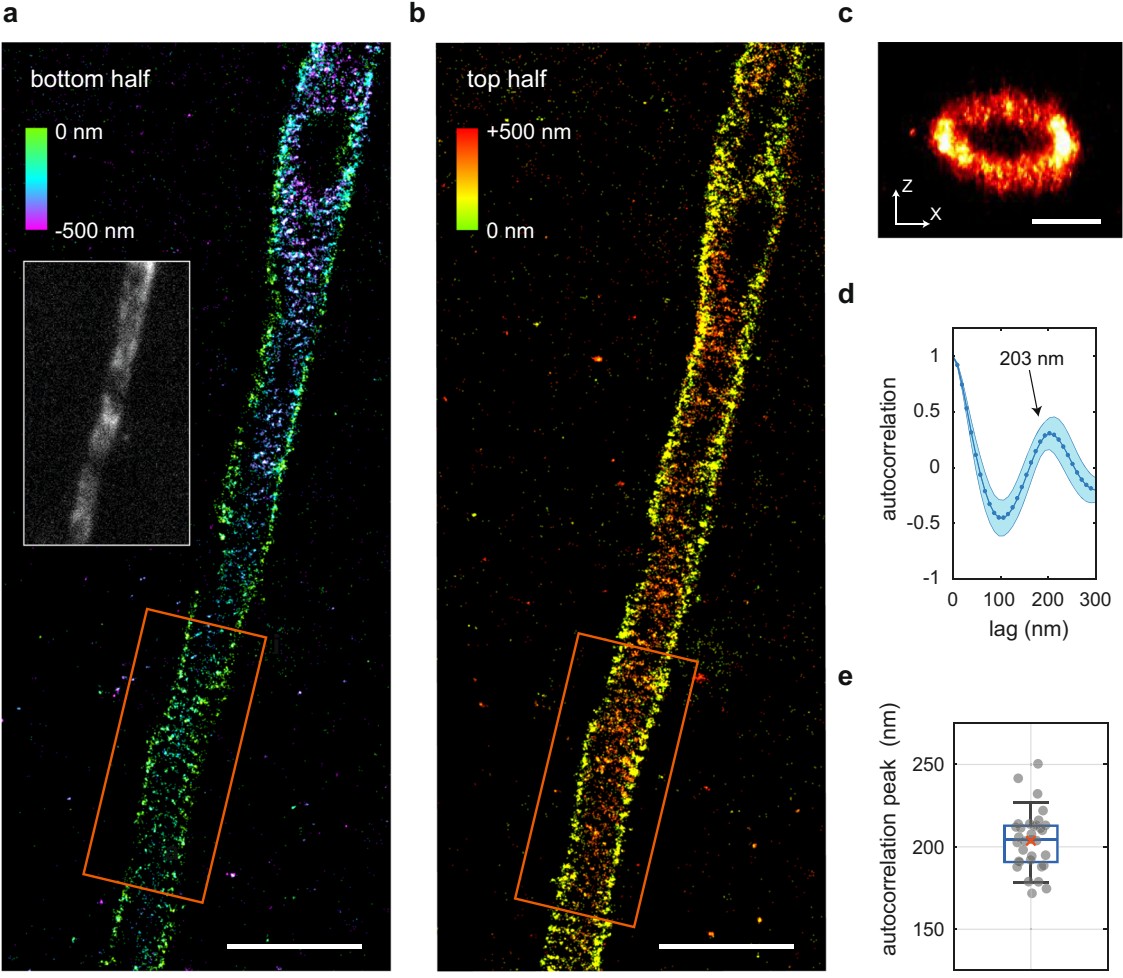

**Fig. 3 Resolving the nanoscale spectrin organization in axons at 50 μm depth. a, b** SMLM reconstruction of the upper half (**a**) and lower half (**b**) of a layer 5 pyramidal neuron AIS stained for βIV-spectrin in a rat brain slice at 50 μm depth. Color encoding indicates the z-position. Scale bars indicates 2 μm. Insert with white rim shows the widefield image prior to aberration correction. The holes in βIV-spectrin pattern are likely sites of synaptic connections. Scale bar indicates 2 μm. **c** z-x cross-section of the rectangular orange area indicated in **a**, **b**. Scale bar indicates 500 nm (both directions). **d** The average autocorrelation (blue line) of 30 line segments in the reconstruction of **a**, **b** and Supplementary Fig. 11 shows a peak at 203 ± 10 nm. Shaded region indicates the standard deviation. **e** Distribution of the estimated autocorrelation peaks of the individual line segments of reconstructions of **a**, **b** and Supplementary Fig. 11. Boxplot indicates 9/91-percentile, 25/75-percentile, and median. Red cross indicates the mean autocorrelation distance. See also Supplementary Fig. 11. Repeated n = 12 times in four distinct samples with similar results. Source data are available as a source data file.

levels ranged between 200 and 300 photons per pixel, resulting in a SBR of ~0.005 (depending on the imaging depth, size of the AIS, and labeling), which is in the range in which REALM can achieve significant correction (see Supplementary Fig. 7).

Beyond 50 μm depth, we found that not the aberration level, but the increase in background fluorescence hampered both aberration correction and localization. In preliminary experiments we noticed that at lower glycerol concentrations (50%), less laser power was required to switch fluorophores to a dark state, which reduced the background levels (to ~50 background photons per pixel) at the expense of more spherical aberration and scattering. The best imaging buffer in tissue might therefore depend on the required imaging depth, the abundance of the staining throughout the sample and the number of fluorophores blinking inside the field of view. To further reduce background, light-sheet based illumination could be considered. In addition, the use of cell-specific in vivo knock-in (KI) approaches in brain tissue, such as ORANGE[20], enables tagging endogenous proteins in only a subset of cells (~1–10%) within the tissue, thereby further reducing background. We anticipate that REALM can enable 80 or 100 μm deep imaging in tissue when combined with

approaches that limit out-of-focus fluorescence by either light-sheet illumination or sparse labeling using an in vivo knock-in approach[20].

REALM can be combined with other z-localization techniques such as PSF engineering[21,22] or self-interference[23]. Our approach also complements the recently introduced In Situ Point Spread function Retrieval (INSPR) localization method, which demonstrated accurate 3D localization in aberrated conditions when imaging below 20 μm depth[24]. Thus, we anticipate that the open source micromanager plug-in REALM that we provide will enable new avenues for SMLM in deep tissue and facilitate correlative functional and nano-structural research.

## Methods

**Set up.** Experiments were performed using a Nikon Ti Eclipse body with a 100× 1.49 NA objective and a quad-band filter cube (containing a ZT405/488/561/640rpc dichroic and ZET405/488/561/640m emission filter, Chroma), to which a MICAO adaptive optics module containing a MIRAO52E (Imagine Optics) deformable mirror was mounted[19] (see Supplementary Fig. 2). Detection was performed with a CMOS camera (Orca Flash v4.0, Hamamatsu). For excitation we used a single mode 647 nm laser (140 mW, LuxX, Omicron) and 405 nm laser

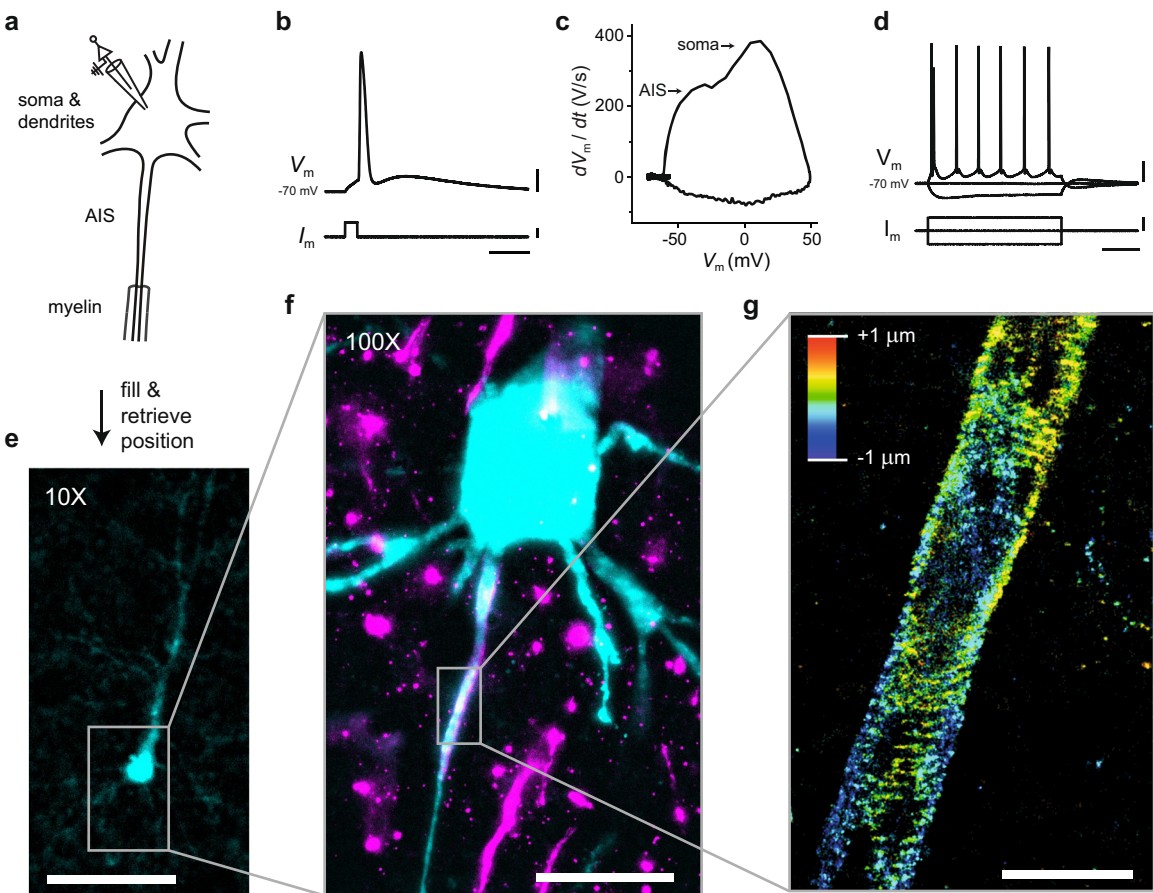

**Fig. 4 Proof-of-principle for combining electrophysiological recordings with 3D SMLM in brain sections. a** Schematic of electrophysiological whole-cell patch clamp recording. **b** An action potential of a layer 5 pyramidal neuron in an acute brain slice patched at 35 µm depth. $V_m$, membrane potential and $I_M$, injected current. Scale bars indicate from top to bottom: 20 mV, 500 pA, and 10 ms. **c** Phase-plane plot ($dV_m /dt$ versus $V_m$) of the action potential shown in **b** showing separate peaks from the AIS and the somatodendritic compartment. **d** The membrane potential ($V_m$) in response to negative and positive current injections ($I_m$). Scale bars indicate from top to bottom: 20 mV, 200 pA, and 200 ms. **e** Widefield image (10×) of the whole-cell recorded and biocytin-filled pyramidal neuron (**b**–**d**) labeled with streptavidin-Alexa488. Scale bar indicates 100 µm. **f** Same as **e** but with a 100× objective. Magenta shows the βIV-spectrin-Alexa647 staining. Scale bar indicates 20 µm. **g** 3D SMLM reconstruction of the βIV-spectrin at the AIS of the same whole-cell patch clamped neuron (**b**–**f**). Single-molecule localizations of four focal planes are merged. Color depicts z-position. Scale bar indicates 2 µm.

(60 mW LuxX, Omicron) which can be used for normal widefield and TIRF illumination.

**Calibration of the DM**. In order to ensure accurate modulation of the Zernike modes, the deformable mirror needs to be properly calibrated. The DM was calibrated by replacing the camera with a Shack-Hartmann sensor (HASO, Imagine Optics) to directly measure the wave-front, using the provided software. A 1-µm bead (TetraSpeck, ThermoFisher, T7282, dilution 1:1000) dried on a coverslip and mounted in glycerol was used as a point source. Mounting in glycerol reduces apodization in the pupil plane caused by super-critical angle fluorescence, ensuring a homogeneously filled pupil. We blocked other beads in the FOV by placing an iris in the intermediate image plane. The wave-front deformation of each actuator was measured with 10 push-pull cycles to obtain the interaction matrix, which was then converted into the Zernike-based control matrix. We perform this calibration once a year.

The calibration was verified using a phase retrieval algorithm[25] in combination with through-focus scans of 175 nm green fluorescent beads (PS-speck, ThermoFisher, P7220, dilution 1:500) mounted in PBS. This revealed that upon startup the shape of the mirror is imperfect due to thermal drift and needs to be corrected (see Supplementary Fig. 3). The inverse estimated Zernike coefficients from the phase retrieval algorithm were subsequently applied by the mirror, after which a second through-focus scan was acquired. This revealed that this approach was able to correct the complete system within 0.1–0.2 rad RMS (15–30 mλ) wave-front error. Prior to each experiment we performed this calibration step to ensure that the system was properly corrected. Next, we modulated individual Zernike modes and acquired through-focus scans of these PSFs. Phase retrieval indicated that all modes up to the 4th order could be accurately modulated with little crosstalk between Zernike modes (see Supplementary Fig. 3).

**DNA-PAINT sample and imaging protocol**. Sample chambers were prepared using double sided tape to create two cavities of 10 µl between a microscope slide and a plasma-cleaned #1.5 high-precision cover glass. Next, 10 µl of a solution of BSA-biotin (1 mg/ml in ultra-pure water MQ, Sigma Aldrich, A8549) was incubated for 5 min after being washed with 50 µl washing buffer (1x PBS containing 10 mM MgCl₂). 10 µl of streptavidin (1 mg/ml in MQ, Sigma Aldrich, 434302) was then flushed in and incubated for 5 min and washed with washing buffer. Next, biotin conjugated to the complementary DNA-strand P1 (1 mg/ml in MQ)[15] was incubated for 5 min and washed away. Finally, washing buffer containing 500 pM of Atto645 conjugated to DNA-strand I1 and green fluorescent beads (PS-speck, ThermoFisher, dilution 1:1000) was flushed in, after which the cavity was sealed with grease and nail polish. The fluorescent beads were used to monitor the stability of the deformable mirror and if needed to correct for thermal drift in between experiments.

For each AO correction, a random aberration configuration consisting of Zernike modes Z2±2, Z3±1, Z3±3, Z40, Z4±2, and Z4±4 (11 modes) was induced by the DM. The amplitudes of these modes were chosen uniformly random and normalized to 0.75 rad RMS. Next, the adaptive optics method was performed (see below for implementation details). Afterwards the residual aberration level $W_{rms}$ was evaluated as

$$W_{rms} = \sqrt{\sum_{j=1}^{11} \left( A_{induced}^{j} - A_{estimated}^{j} \right)^2} \qquad (1)$$

with $A_{induced}^{j}$ the induced known Zernike coefficient of Zernike mode $j$ and the $A_{estimated}^{j}$ the coefficient of Zernike mode $j$ estimated by the AO method. Prior and during the experiment the signal levels and state of the DM were monitored to ensure equal comparison between these methods (see Supplementary Fig. 1 for experimental details).

**Single-molecule acquisition simulations**. The single-molecule acquisitions were simulated using a vector PSF model to capture the full complexity of the aberration configurations[26]. Blinking dynamics play a large role in the variability of the metric value and needed to be incorporated in the simulation. We mimicked these blinking dynamics by introducing a variability in the number of emitters in each frame and in the number of photons each emitter emits. The parameters for these distributions corresponded to the experimental signal levels of the DNA-PAINT sample (see Supplementary Fig. 1). The number of emitters per frame was randomly chosen from a Poisson-distribution (with an average of 13 emitters), and the number of photons of each emitter followed an exponential distribution (with an average of 2500 photons). The emitters were randomly positioned with uniform probability across in the field of view ($400 \times 400$ pixels, 65 nm pixel size) with a uniform background of 20 photons per pixel. Lastly, Poisson noise was added to represent the shot-noise.

The performance of the AO methods was evaluated similar to the procedure described above. Known aberration configurations were induced, followed by AO-based corrections using the different approaches. For comparison with our DNA-PAINT experiment all emitters were placed in focus (z-position = 0 nm) and simulated with a 1.49 NA objective, while for the metric curve and optimization of our method (see Supplementary Figs. 3 and 4) the emitters were uniformly random positioned in the z-direction between ±200 nm at a depth of 20 μm with a 1.35 NA silicon immersion objective with refractive index matching.

**AO methods**. Model-based optimization iteratively corrects Zernike modes by applying a sequence of biases of the Zernike mode that is to be corrected. The metric values of these acquisitions are computed and a metric curve is fitted to find the optimum (see Supplementary Fig. 4). Metric curve fitting was implemented by least-squares fitting with a Gaussian function with an offset (4 fit parameters). The width and center of the fitted Gaussian were constrained to prevent that occasional outliers in the metric values resulted in extreme fit-values. The width was constrained to [0.4, 1] rad and the center to [−0.5, 0.5] rad. This greatly improved the performance of the previously proposed method[11] as the noise-sensitive metric M1 often led to extreme fit-values when not constrained. For Fig. 1, the model-based optimization by Burke et al. was implemented with 11 biases per Zernike mode, whereas REALM used 9 biases per Zernike mode.

The downhill-simplex algorithm uses simplexes (higher dimensional triangles) to find an optimum in the parameter space. We implemented this optimization algorithm via the MATLAB function fmincon with the initial simplex size set to 0.2 rad, which was found to work optimally. Metric M3 was used for all Zernike modes as we did not induce secondary spherical aberration. Therefore, a separate simplex routine with a different metric for primary and secondary spherical aberration as originally proposed[13] could not be implemented. We did not find any reduced correction ability for primary spherical aberration using metric M3. However, we noticed that when correcting more than 4 Zernike modes, the simplex optimization was unable to converge in this larger noisy parameter space. Therefore, optimization was stopped after 300 acquisitions and the state with the best obtained metric value was taken as the estimated correction.

Particle-swarm optimization uses a collection of solutions moving through solution space, where their movement is affected by the individually best solution it found so far, as well as the groups best solution. Particle-swarm optimization was implemented via the MATLAB function particleswarm with a swarm size of 25 as suggested[12] with a maximum of 20 iterations for a maximum of 500 acquisitions. We used an initial swarm spansize of 0.1 rad and a maximum spansize of 0.75 rad. Other settings for particleswarm were set to standard values (InertiaRange, SelfAdjustment, and SocialAdjustment set to 1).

**COS7 staining**. COS−7 cells for Fig. 2a–e were seeded onto 25 mm coverslips. After 24 h, cells were pre-extracted with 0.1% glutaraldehyde and 0.2% Triton-X100 in PEM80 (80 mM Pipes, 1 mM EGTA, 4 mM MgCl₂, pH 6.8) for 1 min. The cells were subsequently fixed with 4% PFA in PEM80 for 10 min. After washing in PBS ($3 \times 5$ min) cells were permeabilized in 0.2% Triton-X100 in PEM80 for 15 min. After washing ($3 \times 5$ min) blocking was performed in 3% BSA in PEM80 for 45 min and incubated overnight with a primary antibody against αTub (mouse IgG1, Sigma Aldrich, B-5-1-2, dilution 1:500). The cells were again washed with PBS ($3 \times 5$ min) and incubated with secondary antibody (goat, anti-Mouse IgG (H + L), AlexaFluor647, Life Technologies, dilution 1:500) for 1 h at RT. The coverslip was then placed on a microscope slide with the cells facing upwards, after which a 50- or 80-μm-thick rat brain section (see below for details) was placed on the coverslip. The surplus of PBS was removed with a tissue and 70 μl of glycerol blinking buffer (see below) was deposited on the slice. Next, a 25 mm #1.5 high-precision coverslip was placed on top of the slice and the assembly was sealed with nail polish. For the blinking buffer, 10 μl of 1 M MEA together with 2.5 μl of 20% glucose, and 1 μl of gloxy buffer (70 mg/ml glucose oxidase, 4 mg/ml catalase in MQ), was mixed with 86 μl of a mixture of 95% glycerol and 5% Tris 20 mM, pH 8.0.

**Slice preparation and βIV-spectrin staining**. All animal experiments were performed in compliance with the European Communities Council Directive 2010/63/EU effective from 1 January 2013. They were evaluated and approved by the

national CCD authority (license AVD8010020172426) and by the Royal Netherlands Academy of Arts and Science (KNAW) animal welfare and ethical guidelines and protocols (IvD NIN 17.21.01 and 19.21.11).

To obtain sections with a fixed thickness (Fig. 2a–f and Supplementary Fig 8 and 10), adult rats were deeply anaesthetized by an i.p. injection of pentobarbital (50 mg/kg) and transcardially perfused with PBS and 4% PFA. The brains were removed and post-fixed in PFA for 24 h after which the tissue was stored in PBS. Coronal sections of 50, 60 or 80 μm thick were cut on a vibratome (VT1200S, Leica Microsystems).

For βIV-spectrin staining (Fig. 3 and 4 and Supplementary Fig. 11 and 13), adult rats were deeply anaesthetized by 3% isoflurane inhalation and decapitated, after which the brains were moved to ice-cold artificial cerebral spinal fluid containing (in mM): 125 NaCl, 3 KCl, 25 glucose, 25 NaHCO₃, 1.25 Na₂H₂PO₄, 1 CaCl₂, 6 MgCl₂, saturated with 95% O₂ and 5% CO₂ (pH 7.4). In total, 300 or 400 μm-thick parasagittal brain sections containing the primary somatosensory cortex were cut on a vibratome (1200S, Leica Microsystems). Following a recovery period at 35 °C for 35–45 min slices were stored at room temperature in the ACSF. For whole-cell filling with biocytin (Fig. 4), the slice was transferred to a customized upright microscope (LNscope, Luigs & Neumann). The microscope bath was perfused with oxygenated (95% O₂, 5% CO₂) ACSF consisting of (in mM): 125 NaCl, 3 KCl, 25 glucose, 25 NaHCO₃, 1.25 Na₂H₂PO₄, 2 CaCl₂, and 1 MgCl₂. Patch pipettes were pulled from borosilicate glass (Harvard Apparatus, Edenbridge, Kent, UK) pulled to an open tip of 3–6 MΩ resistance. The intracellular solution contained (in mM): 130 K-Gluconate, 10 KCl, 4 Mg-ATP, 0.3 Na₂-GTP, 10 HEPES, 10 Na₂-phosphocreatine, and 5 mg ml⁻¹ biocytin (pH 7.25 adjusted with KOH, 280 mOsmol kg⁻¹). An Axopatch 200B (Molecular Devices) was used to obtain whole-cell configuration. The cell was left to fill for 30 min, during which the bridge balance was monitored and stayed below 15 mΩ. Slices were fixed in 4% PFA (20 min) and blocked with 5% NGS and 2% Triton (2 h) before incubation with rabbit anti-βIV-spectrin antibody in blocking buffer (1:1000, 24 h, gift from M. Engelhardt). Slices were washed ($3 \times 15$ min), incubated with goat anti-rabbit Alexa 647 (1:500 or 1:1000, ThermoFisher) and in the case of biocytin filling, with Streptavidin Alexa-488 conjugate (1:500, Invitrogen) and washed again. During all steps, the slices were at room temperature and on a shaker. The slices were stored in PBS (4 °C). Before imaging, slices were incubated for at least 15 min in 95% glycerol and 5% Tris 20 mM after which they were mounted between a microscope slide and #1.5 high-precision coverslip with two 120 μm spacers (Secure-Seal Spacer, Thermofisher, S24735) in the blinking buffer described above.

**SMLM detection, localization, and reconstruction**. The detection, localization and reconstruction was performed with the ImageJ plugin DoM (Detection of Molecules)[27] (see https://github.com/ekatrukha/DoM_Utrecht). DoM detects single molecules events by convolving images with a combination of a Gaussian and Mexican hat kernel. Localization is performed by an unweighted nonlinear 2D Gaussian fit with Levenberg–Marquardt optimization. The detection PSF size was set to 150 nm. Localizations with a width larger than 130% of this size (195 nm) are regarded as false positives. A localization is classified as successful if it is non-false positive and has a positive integrated intensity. SMLM reconstructions were rendered by plotting each molecule as a 2D or 3D Gaussian with standard deviations in each dimension equal to the corresponding localization errors. For astigmatic 3D localization the z-position was estimated from the difference in x- and y-width of the spot, where we corrected for depth-induced loss of astigmatism[19] using a simulated required astigmatism level at each depth (see Supplementary Fig. 13e, f). Drift was corrected by 2D cross-correlation of intermediate reconstructions consisting of 500 or 1000 frames. The FRC resolution was computed by splitting the localizations in two batches every other 500 frames. The corresponding reconstructions where then used the calculate the FRC resolution using the ImageJ Fourier Ring Correlation plugin.

**REALM**. REALM is a free open-source Micro-Manager plugin (github.com/MSiemons/REALM[28]) where the method described in this work is implemented (see Supplementary Fig. 14). It offers a compact and intuitive user interface suitable for non-experts. It currently supports two types of DMs: MIRAO52E (Image Optics) and DMH40-P01 (Thorlabs, see github.com/HohlbeinLab/Thorlabs_DM_Device_Adapter for the device adapter) and we encourage others to build device adapters for other DM manufacturers to interface with REALM. The Fourier transform needed for the metric evaluation is implemented via the 2D Fast Hartley Transform (FHT), where the image is padded with the mode value of the acquisition to a size of $2^n \times 2^n$ (with $n$ in integer), usually resulting in a size of $256 \times 256$ or $512 \times 512$. All aberration corrections for Figs. 2, 3 and 4 are performed with REALM.

**Reporting summary**. Further information on research design is available in the Nature Research Reporting Summary linked to this article.

## Data availability

The data that support the findings of this study are available at FigShare (https://doi.org/10.6084/m9.figshare.c.5412567). Source data are provided with this paper.

## Code availability

The code for REALM is publicly available at github.com/MSiemons/REALM. The custom code for analysis that support the findings of this study are available from the corresponding author upon reasonable request.

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

## Acknowledgements

The authors thank Alard Mosk, Giulia Sereni, and Ivo Vellekoop for useful discussion on aberration in brain tissue, Roderick Tas and Chiung-Yi Chen for help with the DNA-PAINT assays and Audrius Jasaitis (Imagine Optic) for help in controlling the deformable mirror. This work was funded by the Dutch Research Council (NWO) through the FOM program Neurophotonics.

## Author contributions

L.K. and M.S. conceived research. M.S. developed the method presented in this work, performed the simulations, the DNA-PAINT and sandwich assays, analyzed the SMLM data, and made the figures. N.H. prepared the brain sections for the sandwich assay, the βIV stainings, and performed the electrophysiological recording, under supervision of M.K. M.S. and N.H. performed the SMLM imaging on βIV-spectrin. M.S. and L.K. wrote the manuscript with input from N.H. and M.K.; L.K. supervised the project.

## Competing interests

The authors declare no competing interests.
