## [Peer Review File · Nature Communications]

Reviewers' Comments:

Reviewer #1:

Remarks to the Author:

I appreciate that the authors' revision of the manuscript, which has improved its quality. Some of my previous concerns are alleviated in the revised version; however, the lack of experimental results on a realistic thick-tissue fluorescently-labeled sample remains an obstacle.

Reviewer #2:

Remarks to the Author:

I find all rebuttal points reasonable and acceptable. I expect the manuscript to be received warmly by the field and wish you well.

Reviewer #3:

Remarks to the Author:

The Siemons et al manuscript is very interesting. It seems to me that the authors carefully addressed all comments from all three reviewers. The data and the images will highlight the usefulness of the REALM approach for nanoscale molecular imaging in complex tissue preparations. Therefore, this study will be interesting for a broad community of life scientists, and especially for neuroscientists.

I have only very minor comments:

1. Supplementary Figure 14 is an awesome state-of-the-art figure that will open the eyes of numerous neuroscientists for the usefulness of the combination of electrophysiological measurements and nanoscale molecular imaging by using REALM-enhanced SMLM imaging. The workflow in Suppl Fig 14 has already been presented by Dudok et al 2015 Nature Neuroscience for axon terminals, and now the authors extended it for axon initial segments. Because AIS is much larger, its imaging requires thicker preparation hence REALM really makes a difference. I have two suggestions. First, this figure deserves to be a main figure, it could be a nice last figure (Fig 4) as icing on the cake. Second, I would suggest using the presentation workflow logic of Dudok et al 2015 Nature Neuroscience Figure 2, because a reverse order of panels would make the workflow easier to follow for patch-clamp electrophysiologists and anatomists as it would resemble real life experiments. First the researcher collects the physiological information in the slice preparation, and then the morphological and molecular information can be acquired from the very same neuron in a subcellular compartment-specific manner.
2. Page 7, second paragraph "We found without AO blinking events", perhaps "that" is absent.
3. Page 7 "when using the method of Mlodzianoski to 120 nm" perhaps " et al" is absent.
4. Please describe INSPR (in situ point spread function retrieval) in Discussion.

Response to reviewer comments

Reviewer #3 (Remarks to the Author):

The Siemons et al manuscript is very interesting. It seems to me that the authors carefully addressed all comments from all three reviewers. The data and the images will highlight the usefulness of the REALM approach for nanoscale molecular imaging in complex tissue preparations. Therefore, this study will be interesting for a broad community of life scientists, and especially for neuroscientists. I have only very minor comments:

1. Supplementary Figure 14 is an awesome state-of-the-art figure that will open the eyes of numerous neuroscientists for the usefulness of the combination of electrophysiological measurements and nanoscale molecular imaging by using REALM-enhanced SMLM imaging. The workflow in Suppl Fig 14 has already been presented by Dudok et al 2015 Nature Neuroscience for axon terminals, and now the authors extended it for axon initial segments. Because AIS is much larger, its imaging requires thicker preparation hence REALM really makes a difference. I have two suggestions. First, this figure deserves to be a main figure, it could be a nice last figure (Fig 4) as icing on the cake. Second, I would suggest using the presentation workflow logic of Dudok et al 2015 Nature Neuroscience Figure 2, because a reverse order of panels would make the workflow easier to follow for patch-clamp electrophysiologists and anatomists as it would resemble real life experiments. First the researcher collects the physiological information in the slice preparation, and then the morphological and molecular information can be acquired from the very same neuron in a subcellular compartment-specific manner.

➤ We have followed the reviewer's suggested and moved Supplementary Figure 14 to the main figures as the new figure 4. We have also changed the order of the panels, as suggested.

2. Page 7, second paragraph "We found without AO blinking events", perhaps "that" is absent.

➤ Corrected

3. Page 7 "when using the method of Mlodzianoski to 120 nm" perhaps " et al" is absent.

➤ Corrected

4. Please describe INSPR (in situ point spread function retrieval) in Discussion.

➤ We have now described the meaning of INSPR in the discussion.